# Self-Standing Pd-Based Nanostructures for Electrocatalytic CO Oxidation: Do Nanocatalyst Shape and Electrolyte pH Matter?

**DOI:** 10.3390/ijms241411832

**Published:** 2023-07-23

**Authors:** Belal Salah, Adewale K. Ipadeola, Aboubakr M. Abdullah, Alaa Ghanem, Kamel Eid

**Affiliations:** 1Center for Advanced Materials, Qatar University, Doha 2713, Qatar; 2Gas Processing Center (GPC), College of Engineering, Qatar University, Doha 2713, Qatar; 3PVT-Lab, Production Department, Egyptian Petroleum Research Institute, Cairo 11727, Egypt

**Keywords:** porous Pd electrocatalyst, Pd shape effect, alcohol fuel cells, electrocatalytic CO oxidation, Pd nanocube, electrolyte pH effect

## Abstract

Tailoring the shape of Pd nanocrystals is one of the main ways to enhance catalytic activity; however, the effect of shapes and electrolyte pH on carbon monoxide oxidation (CO_Oxid_) is not highlighted enough. This article presents the controlled fabrication of Pd nanocrystals in different morphologies, including Pd nanosponge via the ice-cooling reduction of the Pd precursor using NaBH_4_ solution and Pd nanocube via ascorbic acid reduction at 25 °C. Both Pd nanosponge and Pd nanocube are self-standing and have a high surface area, uniform distribution, and clean surface. The electrocatalytic CO oxidation activity and durability of the Pd nanocube were significantly superior to those of Pd nanosponge and commercial Pd/C in only acidic (H_2_SO_4_) medium and the best among the three media, due to the multiple adsorption active sites, uniform distribution, and high surface area of the nanocube structure. However, Pd nanosponge had enhanced CO_Oxid_ activity and stability in both alkaline (KOH) and neutral (NaHCO_3_) electrolytes than Pd nanocube and Pd/C, attributable to its low Pd-Pd interatomic distance and cleaner surface. The self-standing Pd nanosponge and Pd nanocube were more active than Pd/C in all electrolytes. Mainly, the CO_Oxid_ current density of Pd nanocube in H_2_SO_4_ (5.92 mA/cm^2^) was nearly 3.6 times that in KOH (1.63 mA/cm^2^) and 10.3 times that in NaHCO_3_ (0.578 mA/cm^2^), owing to the greater charge mobility and better electrolyte–electrode interaction, as evidenced by electrochemical impedance spectroscopy (EIS) analysis. Notably, this study confirmed that acidic electrolytes and Pd nanocube are highly preferred for promoting CO_Oxid_ and may open new avenues for precluding CO poisoning in alcohol-based fuel cells.

## 1. Introduction

The unavoidable increase in carbon footprint has a wide range of negative impacts on ecosystems and planet earth, so reducing it is essential [1,2,3,4]. The conversion of gases to value-added products [3,5,6,7,8] and using green energy sources are the main approaches to mitigate carbon footprint [9,10,11,12,13,14]. Among these, fuel cells operating with organic molecules (i.e., ethanol, glucose, ethylene glycol, and formic acid) [15,16,17,18,19,20,21] are among the most promising roadmaps, due to their lower emissions and green resources, but CO poisoning of their anodes is one of the main barriers precluding commercial usage [15,22,23,24]. Therefore, it is important to develop electrocatalysts for efficient CO oxidation (CO_Oxid_) and also to optimize the oxidation process under different conditions that could allow protecting the anodes of organic fuel cells from poisoning by CO. The CO is a highly poisonous gas and causes a lot of hazardous effects on human health and the environment as well, so its conversion to less hazardous CO_2_ gas is important. The CO_Oxid_ enables the mitigation of the hazardous effects and diversify the production sources of CO, with extensive applications in industries for the production of organic chemicals and environmental remediation [25,26,27]. Pd-based electrocatalysts are well known for their outstanding catalytic performance for CO_Oxid_ and other applications, due to their ability to promote CO adsorption and facilitate the dissociation of the water molecules in aqueous electrolytes to generate oxygenated species (i.e., OH) that ease the oxidation process under low potential. Furthermore, they are less prone to carbonaceous species poisoning and can easily be recovered, unlike Pt-based electrocatalysts. The CO_Oxid_ performance of Pd-based electrocatalysts is augmented by tailoring their morphology (i.e., dimensional, surface features, and porosity), compositions (i.e., alloying), and size, in addition to using supports (i.e., metal oxide and carbon) [28,29,30,31,32,33].

To this end, tailoring the morphology of Pd nanocrystals results in tuning the Pd-Pd interatomic distance and d-band center, which enhance the electrocatalytic performance, driven by strengthening the adsorption of reactants (i.e., OH and O_2_), besides easing their activation/dissociation and weakening the desorption of products (i.e., CO_2_) [34,35,36,37,38,39]. This is based on using a wide range of methods, such as template-assisted [40], solvothermal/hydrothermal [41], galvanic replacement [42], seed-mediated [43], and aqueous solution reduction [44,45,46] For example, the CO_Oxid_ current density of Pd/ZIF-67/C (4.2 mA/cm^2^) was superior to that of Pt/C (1.08 mA/cm^2^) by 3.88 folds and Pd/C (0.95 mA/cm^2^) by 4.42 folds in H_2_SO_4_ electrolyte, owing to the effect of the ZIF-67/C support with porous structure, great surface area, and Co-N_x_ sites [47]. Likewise, the CO_Oxid_ current density of PdNiO-CeO_2_/onion-like carbon (2.50 mA/cm^2^) in H_2_SO_4_ electrolyte was 1.74, 3.73, and 2.63 times those of PdNiO/OLC, PdNiO-CeO_2_, and Pd/C, respectively, in addition to it having higher durability, due to the effect of coupling two supports [48]. Pd/Ni-MOF/PC had promoted CO_Oxid_ current density (4.71 mA/cm^2^) in H_2_SO_4_ medium that outperformed Pd/Ni-MOF/C (1.38 mA/cm^2^) and Pd/C (0.95 mA cm^2^) by factors of 3.41 and 4.96, owing to the effect of the Ni-MOF/PC support with high porosity, surface area, and Ni-N_x_ [49]. Pd nanoparticles supported on Ti_3_C_2_T_x_ MXene exhibited enhanced CO_Oxid_ activity compared to Pd/C, due to the interaction of MXene support [50]. Previous studies focused on the CO_Oxid_ performance of Pd nanoparticles supported on various supports (i.e., carbon and metal oxides) without emphasizing the activity of pristine Pd nanoparticles. Meanwhile, the electrocatalytic CO_Oxid_ of self-sanding Pd nanocatalysts with much focus on the effect of shape and electrolyte pH are not highlighted enough compared with other applications [46].

To this end, we attempted to emphasize and solve these issues via the rational preparation of Pd nanosponge and Pd nanocube whose electrocatalytic CO_Oxid_ performances are compared in different electrolytes, including alkaline (KOH, pH = 12), acidic (H_2_SO_4_, pH = 2), and neutral (NaHCO_3_, pH = 7.4), besides an excellent regeneration of the Pd nanocrystals. This is to underline the role of Pd shapes and electrolyte type/pH on the CO_Oxid_ activity of Pd. The formation process of Pd nanocrystals is driven by the direct reduction of the Pd precursor by ascorbic acid under sonication, while Pd nanosponge is formed through ice-cooling reduction with NaBH_4_ solution. Both Pd nanosponge and Pd nanocube possess many advantages, such as being support-free and having a great electrochemical active surface area, uniform distribution, and clean surface. This is in addition to the simplicity, greenness (i.e., no organic solvents or surfactants), and speed (only 5 min) of the preparation approaches. The electrocatalytic CO_Oxid_ activity and durability of Pd nanocube were tested relative to Pd nanosponge and commercial Pd/C nanosphere in different electrolytes at room temperature. EIS measurements were conducted to obtain more insights into the charge transfer resistance and electrolyte–electrode interaction on Pd morphologies over a wide pH range during CO_Oxid_.

## 2. Results and Discussion

Figure 1a describes the synthesis pathways of Pd nanocube nanostructures via the chemical reduction of K_2_PdCl_4_ by ascorbic acid to induce the nucleation of Pd^2+^ to Pd^0^ and subsequent growth, where PVP is adsorbed on Pd nuclei to direct the formation of cubic-like morphology. Meanwhile, Pd nanosponge were formed via the strong reduction power of NaBH_4_, which induces the burst nucleation to form massive Pd crystallites, which tend to coalesce and reduce their thermodynamic instability, leading to a coalescence growth mechanism [15,17]. The SEM image manifests the formation of Pd nanocube (Figure 1b) and Pd nanosponge (Figure 1c) in high yield. The Pd nanosponge is composed of small nanoparticles assembled in a 3D porous sponge structure. The porosity may result from the gases released in situ (H_2_ and diborane B_2_H_6_) during ice-cooling reduction by NaBH_4_ [15,17]. The TEM images of Pd nanocube display the cubic-like morphology, which is uniformly distributed with an average size of (10.0 ± 0.6 nm) (Figure 1d); meanwhile, the Pd nanosponge comprises small aggregated Pd nanoparticles (8.0 ± 0.4 nm) assembled in a chain-like shape that gather together to form a sponge-like shape (Figure 1e). The noticed aggregation of Pd nanosponge is due to the coalescence growth [15,17]. The spherical-like morphology of Pd/C is revealed in Appendix A.

The HRTEM image of a randomly selected Pd nanocube showed a cubic-like shape with obvious lattice fringes extended in different directions, implying non-epitaxial growth (Figure 1f). The lattice fringes are clear without any defects, indicating the purity of Pd nanocube, and the resolved interplanar distances are assigned to {111}, {200}, {220}, and {3111} facets of the face-centered cubic (*fcc*) crystal structure of Pd, which conformed with the Fourier filter images (FFT) (Figure 1g). The high-index facets may result from the ability of the facet-capping agent PVP to direct the crystal growth. Meanwhile, these facets are usually obtained in polyhedron Pd shapes (i.e., nanocube or octahedral), which is highly favored in promoting the catalytic properties of Pd [15,17]. The HRTEM of the Pd nanosponge demonstrates the aggregated nanoparticles with lattice fringes attributed to {111} facet of *fcc* Pd (Figure 1h). The SAED patterns of Pd nanocube and Pd nanosponge display the diffraction spots of {111}, {200}, {220}, {3111} and {222} of *fcc* Pd (Figure 1i,j), as commonly detected in Pd-based nanostructures [50,51,52].

The EDS analysis of Pd nanocube and Pd nanosponge confirms the presence of only pure Pd atoms (100%) without any impurities (Figure 1k), due to the high reduction ability of both ascorbic acid and NaBH_4_ towards the Pd precursor with high reduction potential (Pd^2+^/Pd^0^ (0.92 V)). On the other hand, the TEM image of Pd/C shows the formation of Pd nanosphere with an average diameter of (4.0 ± 1.0 nm) distributed over carbon sheets (Appendix A).

The diffraction patterns of Pd nanocube and Pd nanosponge display the *fcc* of Pd, which are similar to those of commercial Pd/C catalysts with the most exposed {111} facets (Figure 2a) [50,53]. No catalysts showed any peaks of the Pd oxide phase, owing to the strong reduction ability of the reducing agents (NaBH_4_ and ascorbic acid) [15,17]. Notably, the XRD diffraction patterns of Pd nanocube and Pd nanosponge are slightly shifted negatively compared with those of Pd/C, which indicates the lattice expansion of Pd nanocube and decreased Pd-Pd interatomic distance of Pd nanocube and Pd nanosponge compared to Pd/C [15,17]. This is further evidenced by the lower lattice parameters of Pd nanocube (3.20 Å) and Pd nanosponge (3.18 Å) relative to Pd/C (3.89 Å). Modulating the Pd-Pd interatomic distance of Pd is beneficial for tailoring the electronic properties and increasing the catalytic activity [15,17]. The calculated crystallite sizes at {111} facet are 3.2 nm for Pd nanocube and 2.5 nm for Pd nanosponge, as calculated using the Scherrer equation.

The XPS spectra of the Pd nanocube and Pd nanosponge reveal the core-level of Pd 3d, but with a noticeable left-shift in the binding energy for Pd nanocube relative to Pd nanosponge and pure Pd (Figure 2b). The lowered binding energy serves as an indication of the increased d-band center of Pd with respect to the Fermi level, which is one of the decisive roadmaps for improving the electronic structure and catalytic properties of Pd [15]. This is driven by boosting the adsorption of reactants (CO/O_2_) along with facilitating their activation and dissociation during CO oxidation [47,48,49]. The fitting of Pd 3d spectra in Pd nanocube and Pd nanosponge show the main peaks assigned to zero-valent Pd (Pd^0^ 3d_5/2_ and Pd^0^ 3d_3/2_), besides a minor phase of Pd^2+^ and Pd^4+^ (Figure 2c,d) (Appendix A). The ratio of Pd^0^ to Pd^2+^ in Pd nanocube (1.2) was lower than that in Pd nanosponge (1.4). The presence of Pd^2+^ and Pd^4+^ is preferred for promoting the activation/dissociation of water in electrolytes and accelerating the CO_Oxid_ kinetics [15,47,48,49].

To investigate the effect of Pd shape and electrolyte pH, the electrochemical CO_Oxid_ activity and stability were tested on Pd nanocube, Pd nanosponge, and Pd/C nanosphere in H_2_SO_4_, KOH, and NaHCO_3_. The CV curves tested in N_2_-saturated aqueous H_2_SO_4_ solution on all electrocatalysts depict typical voltammogram features of Pd-based catalysts, and it can be observed that the peaks assigned to the reduction of Pd-O on Pd nanocube and Pd nanosponge were significantly higher and left-shifted relative to that in Pd/C, which is attributed to the increased d-band center (Figure 3a) [47,48,49]. This implies the superior ability of Pd nanocube and Pd nanosponge to easily form oxygenated species (i.e., OH and O_2_), which can accelerate the CO_Oxid_ kinetics under low potential [47,48,49]. In addition, the hydrogen under-potential deposition (H-UPD) area of Pd nanocube and Pd nanosponge were larger than that of Pd/C, indicating their greater ECSA. Thereby, the ECSA of Pd nanocube (22.88 m^2^/g) and Pd nanosponge (23.63 m^2^/g) were higher than that of Pd/C (11.1 m^2^/g), due to their multidimensional (i.e., 3D) structures (Table 1).

In CO-saturated H_2_SO_4_ solution, the CO_Oxid_ voltammogram features include a sharp peak in the forward direction at a higher potential (0.55–0.8 V) with an obvious anodic current (*I*_Anode_) and another peak in the backward direction with a clear cathodic current (*I*_Cathode_) at (0.5–0.45 V), showed with red arrow, but with noticeably higher activity on Pd nanocube than its counterparts (Figure 3b). Regarding this, the *I*_Anode_ of Pd nanocube (5.92 mA/cm^2^) was 1.48 times that of Pd nanosponge (4 mAcm^−2^) and 3.72 times that of Pd/C (1.59 mAcm^−2^), owing to the multiple accessible active sites of nanocube and its high-index facets (Figure 3b), which enhance the CO/O_2_ adsorption, followed by their promoted dissociation to allow complete oxidation under low potential [47,48,49]. Thereby, the amount of CO adsorbed on Pd nanocube was (930 μC), Pd nanosponge (955.5 μC), and Pd/C (265 μC) in 0.1 M H_2_SO_4_, as obtained from integrated CO oxidation charges on the catalysts between (0.55 and 0.8 V). Thus, the CO_Oxid_ mass (specific) activities of Pd nanocube (19.85 mA/mg (0.087 mA/cm^2^)) were greater than those of Pd nanosponge (14.13 mA/mg (0.059 mA/cm^2^)) and Pd/C (18.72 mA/mg (0.16 mA/cm^2^)) (Table 1). This serves as an indication for the maximized utilization of the Pd atoms and active sites of Pd nanocube during CO_Oxid_ in H_2_SO_4_ compared to Pd nanosponge and Pd/C [47,48,49]. In addition, both Pd nanocube and Pd nanosponge had reduced CO_Oixd_ potential (*E*_Oxid_) and onset potential (*E*_Onset_) at an earlier potential than that of Pd/C. This is seen in the lower *E*_Oxid_ (*E*_Onset_) of Pd nanocube (0.76 V (0.36 V)) and Pd nanosponge (0.72 V (0.41 V)) relative to Pd/C (0.88 V (0.75 V)) (Figure 3c). This suggests that there is better oxidative removal of carbonaceous intermediates on Pd nanocube and Pd nanosponge, which leads to accelerating the CO_Oxid_ kinetics and circumvents blocking the active sites of Pd [47,48,49].

The CO_Oxid_ activity of Pd nanocube was superior to previous reports of Pd-based electrocatalysts measured under similar conditions (Appendix A). The CV curves measured at varying scan rates (*υ*) on Pd nanocube, Pd nanosponge, and Pd/C show a steady increase in the *I*_Anode_ with increasing *υ* (Figure 3d–f). In addition, the relationship between *υ*^1/2^ and *I*_Anode_ was linear, which suggests a diffusion-controlled CO_Oxid_ process on Pd nanocube, Pd nanosponge, and Pd/C. However, the line slopes of Pd nanosponge (0.98 ± 0.05) and Pd nanocube (1.15 ± 0.09) were higher than that of Pd/C (0.34 ± 0.01) with regression coefficients (0.994–0.998) (Figure 3d–f), signifying better transportation kinetics on Pd nanocube and Pd nanosponge, which is plausibly attributed to the shape effect [15,47,48,49].

Pd nanocube displays greater durability than Pd nanosponge and Pd/C over 3 h in H_2_SO_4_ electrolyte, as confirmed by the CA test (Figure 4a). The CV curves measured after CA indicate that all catalysts retain their initial CO_Oxid_ voltammogram features, but with a higher stability on Pd nanocube than Pd nanosponge and Pd/C (Figure 4b–d). This is evidenced by the ability of Pd nanocube to maintain about 86.7% of the initial value compared to Pd nanosponge (77.5%) and Pd/C (64.5%). The superior durability of Pd nanocube is attributed to the multidimensional shape with multiple active sites, high-index facets, and modulated Pd-Pd distance, which ease the desorption of poisoning intermediates during CO_Oxid_, while porous Pd nanosponge allows quick diffusion of reactants and tolerates the adsorption of intermediates [15,47,48,49]. Meanwhile, the loss of CO_Oxid_ activity in Pd/C is due to the high feasibility of Pd nanoparticles to detach and aggregate during CO_Oxid_, so Pd nanocube and Pd nanosponge maintain their active sites. Thereby, the ECSA of Pd nanocube shows only a 6.1% degradation compared to Pd nanosponge (8.2%) and Pd/C (15.8%) (Appendix A). This is verified by the TEM images recorded after the durability tests, which show the morphological stability of Pd nanocube and Pd nanosponge, but Pd/C shows an aggregation of Pd nanoparticles (Appendix A).

The EIS tests show Nyquist plots that exhibit semicircle lines, but with a lesser diameter for Pd nanocube and Pd nanosponge than Pd/C, implying lower charge transfer resistance and a better electrolyte–electrode interface on Pd nanocube and Pd nanosponge (Figure 4e). The validity of this observation is further supported by fitting and analyzing the EIS data using the Voigt electrical equivalent circuit (EEC, Appendix A). Intriguingly, Pd nanocube displayed a lower electrolyte resistance (*R*_s_) and charge transfer resistance (*R*_ct_) than those of Pd nanosponge and Pd/C (Appendix A), demonstrating the higher ionic conductivity and charge mobility on Pd nanocube [15,47,48,49]. This is verified by the power law of constant-phase element (CPE) impedance (*Z*_CPE_ = 1/(Q(jω)**^α^**) with ideality factor (α), which exhibited a higher CPE on Pd nanocube (55.25 μS.s^(1−^**^α^**^)^) than Pd nanosponge (42 μS·s^(1−^**^α^**^)^) and Pd/C (25.30 μS·s^(1−^**^α^**^)^), indicating the lower charge/mass transfer resistance on the Pd nanocube. This is also shown in the lower α on Pd nanocube (Appendix A). This is further corroborated by the Bode plots (Figure 4f), which disclose a lower total impedance in the low-frequency region on Pd nanocube relative to the Pd nanosponge and Pd/C [47,48,49]. Additionally, Pd nanocube and Pd nanosponge had a lower phase angle (68.6–76.3°) than Pd/C (77.0°), signifying the superior diffusion of carbonaceous species intermediates on Pd nanocube during CO_Oxid_. Moreover, the Pd nanocrystals and PdC were regenerated after CO oxidation in 0.1 M H_2_SO_4_ electrolyte by purging the solution with N_2_ for 1 h to remove the adsorbed CO and/or CO_2_ on the catalysts’ active sites (Appendix A). The Pd nanocube and Pd nanosponge showed excellent regeneration after the CO gas was switched to N_2_, evidenced by the absence of CO oxidation compared to Pd/C.

The CO_Oxid_ mechanism on Pd nanocube could plausibly follow the Langmuir–Hinshelwood mechanism, as shown in Equations (1)–(3). This involves the co-adsorption of CO (CO_ads_) and hydroxyl OH adsorption (OH_ads_) on the Pd surface. The dissociation of water on the Pd surface (H_2_O ↔ H^+^ + OH^−^) in the aqueous H_2_SO_4_ medium generates OH species needed for promoting the oxidation of CO_ads_ on the Pd surface at lower potential [19]. Finally, the CO_2_ formed is promptly desorbed from the surface of Pd nanocube [19].
(1)$$\mathrm{Pd}\;\mathrm{nanocube}+\mathrm{CO}\;\to \;\mathrm{Pd}\;\mathrm{nanocube}-{\mathrm{CO}}_{\mathrm{ads}}$$
(2)$$\mathrm{Pd}\;\mathrm{nanocube}+H_{2}O\;\to \;\mathrm{Pd}\;\mathrm{nanocube}-{\mathrm{OH}}_{\mathrm{ads}}+H^{+}+e^{-}$$
(3)$$\mathrm{Pd}\;\mathrm{nanocube}-{\mathrm{OH}}_{\mathrm{ads}}\;+\mathrm{Pd}\;\mathrm{nanocube}-{\mathrm{CO}}_{\mathrm{ads}}\to \;{\mathrm{CO}}_{2}\;+2\;\mathrm{Pd}\;\mathrm{nanocube}+H^{+}+e^{-}$$

The CV curves tested in N_2_-saturated KOH display Pd voltammogram features with a higher H-UPD area on Pd nanocube and Pd nanosponge than Pd/C catalysts, implying their higher ECSA. Therefore, the ECSA of Pd nanosponge (15.1 m^2^/g) was superior to Pd nanocube (2.85 m^2^/g) and Pd/C (2.77 m^2^/g) (Table 1). Notably, the reduction in the Pd-O peak on Pd nanocube and Pd nanosponge was shifted to higher potential relative to Pd/C, indicating the sluggish formation of oxygenated species on Pd nanosponge and Pd nanocube; however, the peak area of Pd-O reduction was significantly greater than Pd/C, implying their higher active sites (Figure 5a). This trend is different from that found in the measurements in H_2_SO_4_ electrolyte, which is in line with previous reports, which showed alteration of the oxidation behavior in different electrolytes [15,47,48,49].

The CO_Oxid_ voltammogram features were obtained on all electrocatalysts, but with a greater *I*_Anode_ on Pd nanosponge (3.2 mA/cm^2^) by 1.96 times that of Pd nanocube (1.63 mA/cm^2^) and 2.16 times that of Pd/C (1.48 mA/cm^2^) (Figure 5b). The CO_Oxid_ charges calculated from an integrated peak at (0.55–0.9 V) in the forward scan were found to be (602 μC) on Pd nanosponge, (110 μC) on Pd nanocube, and (99.54 μC) on Pd/C, indicating greater (CO/OH) adsorption on Pd nanosponge and Pd nanocube than Pd/C. Nevertheless, the ease of formation of OH species on Pd/C allows CO_Oxid_ at earlier oxidation and onset potential than Pd nanosponge and Pd nanocube, as shown in the LSV test (Figure 5c).

The CV curves obtained at different *υ* showed an increase in the *I*_Anode_ at high *υ*, in addition to a linear relationship between *υ*^1/2^ and *I*_Anode_ on all electrocatalysts, but with a greater line slope on Pd nanosponge (0.539) than Pd nanocube (0.31) and Pd/C (0.29) (Figure 5d–f). This implies the CO_Oxid_ process is a diffusion-controlled process, but with better diffusion on Pd nanosponge and Pd nanocube than Pd/C, attributed to the shape effect as noted previously in the case of Pd nanoparticles [20,54].

The slower current attenuation and lower degradation in the current density on Pd nanosponge than Pd nanocube and Pd/C as shown in the CA curves after 3 h imply the great stability of Pd nanosponge (Figure 6a). The CV curves tested after the CA tests demonstrated the voltammogram characteristics, but with higher retained *I_Anode_* on Pd nanosponge (78%) than Pd nanocube (39%) and Pd/C (26%) (Figure 6b–d). This is due to the higher ECSA of Pd nanosponge, which only degraded by (20%), relative to Pd nanocube (45%) and Pd/C (26%) (Appendix A) [15,47,48,49].

Nyquist plots with semicircle lines were obtained for all electrocatalysts, but with a smaller diameter on Pd nanosponge and Pd nanocube than Pd/C, which indicates the lower charge transfer resistance and improved electrolyte–electrode interface on Pd nanocube and Pd nanosponge (Figure 6e) [15,47,48,49]. This was proven by fitting and analyzing the EIS data using EEC (Appendix A), which demonstrated the lower *R*_s_ and *R*_ct_ on Pd nanosponge and Pd nanocube than Pd/C (Appendix A). This reveals the superior ionic conductivity and lower charge transfer resistance on Pd nanosponge and Pd nanocube than Pd/C. This is also proven by the *Z*_CPE_ law, which infers greater CPE and α on Pd nanosponge and Pd nanocube than Pd/C (Appendix A). This serves as an indication of the higher charge mobility on the Pd nanocube. This is also revealed in the lower impedance and phase angle at the low-frequency region on Pd nanosponge and Pd nanocube than Pd/C, as shown in the Bode plots (Figure 6f), which depict the faster diffusion of carbonaceous species intermediates on Pd nanosponge and Pd nanocube during CO_Oxid_ [15,47,48,49].

The CV curves tested in 0.5 M NaHCO_3_ with N_2_ saturation showed the voltammogram of a typical Pd, but with a higher H-UPD area and reduction in the Pd-O peak on Pd nanosponge compared to Pd nanocube and Pd/C catalyst, indicating their greater active sites and surface area (Appendix A). Thereby, the ECSA were 10.11, 6.23, and 3.16 m^2^/g on Pd nanosponge, Pd nanocube, and Pd/C, respectively (Table 1). The noticeable shift in the potential of the Pd-O peak on Pd nanosponge relative to Pd nanocube and Pd/C indicates their ability to accelerate the generation of oxygenated species at lower potential [15,47,48,49]. In an aqueous NaHCO_3_ electrolyte saturated with CO, the voltammogram characteristics assigned to the CO_Oxid_ were observed, but with an obvious higher *I*_Anode_ on Pd nanosponge (3.2 mA/cm^2^) than Pd nanocube (1.63 mA/cm^2^) and Pd/C (1.48 mA/cm^2^) (Appendix A). The integration of the CO_Oxid_ charges from the peak at (0.55–0.9V) in the positive potential direction on Pd nanosponge (602 μC) was substantially greater than on Pd nanocube (110 μC) and Pd/C (99.54 μC). This suggests higher (CO/OH) adsorption on Pd nanocube and Pd nanosponge than Pd/C under low applied potential, as revealed in the LSV, which showed the occurrence of CO_Oxid_ at earlier oxidation and onset potential on Pd nanosponge and Pd nanocube than Pd/C (Appendix A).

The *I*_Anode_ increased with increasing *υ* on the electrocatalysts along with a noticeable linear relationship between *υ*^1/2^ and *I*_Anode_, but an obviously higher line slope on Pd nanosponge (0.28) than Pd nanocube (0.13) and Pd/C (0.11) (Appendix A). This plausibly serves as proof for the diffusion-controlled process of CO_Oxid_ on the electrocatalysts, but with quicker diffusion on Pd nanosponge and Pd nanocube than Pd/C, resulting from the shape effect [20]. The CA displays the better durability of Pd nanosponge than Pd nanocube and Pd/C (Appendix A), as also shown in the CO_Oxid_ voltammogram after CA tests. Pd nanosponge maintained nearly 78% of its *I*_Anode_ relative to Pd nanocube (39%) and Pd/C (26%) (Appendix A). Additionally, the ECSA of Pd nanosponge, Pd nanocube, and Pd/C were maintained at 82, 72, and 67%, respectively (Appendix A). The EIS measurements display the typical Nyquist plots with semicircle lines with a lesser diameter on Pd nanosponge and Pd nanocube than Pd/C, designating the lower charge transfer resistance and enhanced electrolyte–electrode interaction on Pd nanocube and Pd nanosponge (Appendix A). Fitting and analyzing the EIS data using EEC verified the lower *R*_s_ and *R*_ct_ on Pd nanosponge and Pd nanocube than Pd/C (Appendix A), disclosing their greater ionic conductivity and lower charge transfer resistance. The *Z*_CPE_ law infers a bigger CPE and α on Pd nanosponge and Pd nanocube than Pd/C (Appendix A), indicating their quicker charge mobility. The smaller impedance and phase angle in the low-frequency region on Pd nanosponge and Pd nanocube than Pd/C, as shown in the Bode plots, depict their ability to accelerate the diffusion of carbonaceous intermediates (Appendix A) [15,47,48,49].

These results show the substantial effect of the Pd shape and electrolyte pH on promoting the CO_Oxid_ performance of Pd-based catalysts. However, the self-standing Pd nanocube and Pd nanosponge outperformed Pd/C catalysts in the three electrolytes. This originated from the multidimensional structure with multiple surface corners, active sites, and high surface area, which facilitate the adsorption and diffusion of the reactants, ease the desorption of intermediates and products, and maximize the utilization of Pd active sites during CO_Oxid_ [15,47,48,49].

## 3. Materials and Methods

### 3.1. Materials/Chemicals

Potassium tetrachloropalladate (II) (K_2_PdCl_4_, ≥99%), sodium borohydride (NaBH_4_, 99%), L-ascorbic acid (≥99%), polyvinylpyrrolidone (PVP), potassium hydroxide (KOH, ≥99.95%), sodium bicarbonate (NaHCO_3_, ≥99.7%), sulfuric acid (H_2_SO_4_, ≥99.99%), and commercial Pd/C (20 wt.%) were purchased from Sigma-Aldrich (Chemie GmbH, Munich, Germany).

### 3.2. Preparation of Pd Nanocube

Pd nanocube was prepared through a method previously described with minor modifications [15,55]. The procedure involved mixing an aqueous solution of K_2_PdCl_4_ (15 mM, 10 mL) and PVP (13.0 mg) under sonication, followed by the rapid addition of L-ascorbic acid (0.1 M, 1.0 mL) and left for 5 min. The resultant Pd nanocube was isolated via three sequential centrifugation at 7000 rpm for 10 min and washed with deionized H_2_O.

### 3.3. Preparation of Pd Nanosponge

The Pd nanosponge was synthesized through a method previously reported [15,55], including the prompt addition of an aqueous solution of NaBH_4_ (0.1 M, 2.0 mL) to an aqueous solution of K_2_PdCl_4_ (15 mM, 20 mL) in an ice bath under sonication for 3 min at 0 °C. The resulting opaque black precipitate was dispersed in deionized H_2_O under sonication at 25 °C, followed by centrifugation at 7000 rpm for 5 min and washing with deionized water.

### 3.4. Material Characterization

A scanning electron microscope (SEM) equipped with an energy dispersive X-Ray analyzer (EDX) (Hitachi S-4800, Hitachi, Tokyo, Japan) and a transmission electron microscope (TEM) (TecnaiG220, FEI, Hillsboro, OR, USA) were used for shape and composition analysis. The X-ray photoelectron spectroscopy (XPS) was conducted using an Ultra DLD XPS Kratos (Manchester, UK), while X-ray diffraction (XRD) was performed on \n X’Pert-Pro MPD (PANalytical Co., Almelo, The Netherlands). The inductively coupled plasma optical emission spectrometry (ICP-OES) was carried out using an Agilent 5800, Santa Clara, CA, USA).

### 3.5. Electrochemical CO_Oxid_ Measurements

The electrocatalytic CO_Oxid_ was evaluated with cyclic voltammetry (CV), linear sweep voltammetry (LSV), impedance spectroscopy (EIS), and chronoamperometry (CA) tests on a Gamry potentiostat (Reference 3000, Gamry Co., Warminster, PA, USA) using a three-electrode cell system consisting of a Pt wire, Ag/AgCl, and glassy carbon electrodes (GCEs) (∅5 mm × 1 mm) as the counter, reference, and working electrodes, respectively. The working electrodes were polished with alumina powder of different sizes and rinsed with ethanol and deionized water several times under sonication, then covered with the catalyst inks. The catalyst ink (2 mg) was dissolved in an aqueous solution of 1 mL ethanol/H_2_O/Nafion (3/1/0.5 *v*/*v*) under sonication for 10 min. The catalyst ink was deposited onto the working electrodes and left to dry in an oven under vacuum at 50 °C for 1 h. The catalyst loading on the working electrode was approximately 0.2831 ± 0.0001 μg_Pd_ based on ICP-OES measurements. Before the measurements, a CV test was conducted on each electrolyte under continuous purging of N_2_ at 200 mV/s for 100 cycles to remove any impurities, followed by measurement at 50 mV/s for 3 cycles to determine the electrochemical active surface area (ECSA). The ECSA was calculated using coulombic charge (Q), Pd loading (m) on the working electrode, and coulombic constant for monolayer of Pd (0.424 mC/cm^2^) from Equation (4).
(4)$$\mathrm{ECSA}=\frac{Q}{Sxm}$$

After that, the electrodes were transferred to another cell with fresh electrolytes and exposed to continuous CO purging to measure the CO_Oxid_. The EIS measurements were conducted on the catalysts under the CO oxidation potential of each catalyst under a frequency range (0.1 Hz to 100 kHz) with an AC voltage amplitude of 5 mV at open circuit potential in different electrolytes. The Voigt electrical equivalent circuit (EEC) was used for the fitting and analysis of the EIS measurements.

## 4. Conclusions

In brief, this article emphasizes the effect of Pd shape and electrolyte pH on CO_Oxid_ activity and durability, comprising the synthesis of Pd nanosponge through ice-cooling reduction of the Pd precursor by NaBH_4_ solution and Pd nanocube through ascorbic acid reduction at 25 °C in the presence of PVP. The surface and bulk analysis reveal the formation of self-standing Pd nanosponge and Pd nanocube with controlled morphology, good distribution, and great ECSA, which endowed the Pd nanosponge and Pd nanocube with enhanced CO_Oxid_ than that of Pd/C in different electrolytes, including alkaline (KOH, pH = 12), acidic (HClO_4_, pH = 2), and neutral (NaHCO_3_, pH = 7.4). The CO_Oxid_ activity and stability of Pd nanocube were expressively higher than those of Pd nanosponge and commercial Pd/C nanosphere in HClO_4_ only, but Pd nanosponge exhibited high CO_Oxid_ activity and stability in both KOH and NaHCO_3_ electrolytes, which originated from the multiple adsorption active sites, uniform size distribution, and greater ECSA of Pd nanocube morphology, in only acid medium, whereas the enhanced CO_Oxid_ activity and durability of Pd nanosponge in both alkaline and neutral media is attributable to its low Pd-Pd interatomic distance and cleaner surface. Pd nanocube had enhanced CO_Oxid_ current density in HClO_4_ that reached 5.92 mA/cm^2,^, which was 3.61 and 10.30 times higher than those in KOH (1.63 mA/cm^2^) and in NaHCO_3_ (0.58 mA/cm^2^) electrolytes, respectively. In addition, the Pd atomic utilization was higher in the HClO_4_ electrolyte, as shown in the greater CO_Oxid_ mass activity of Pd nanocube (39.7 mA/mg_Pd_) in HClO_4_ than in KOH (11.5 mA/mg_Pd_) and in NaHCO_3_ (4.08 mA/mg_Pd_). The EIS analysis demonstrates the lower charge transfer resistance and better electrolyte–electrode interaction on Pd nanocube in HClO_4_ than those in KOH and in NaHCO_3_. This study proved that Pd nanocube in acidic electrolyte is favorable for boosting CO_Oxid_ performance, which can provide guidelines for impeding the CO poisoning of anodes in alcohol-based fuel cells.

## Figures and Tables

**Figure 1 ijms-24-11832-f001:**
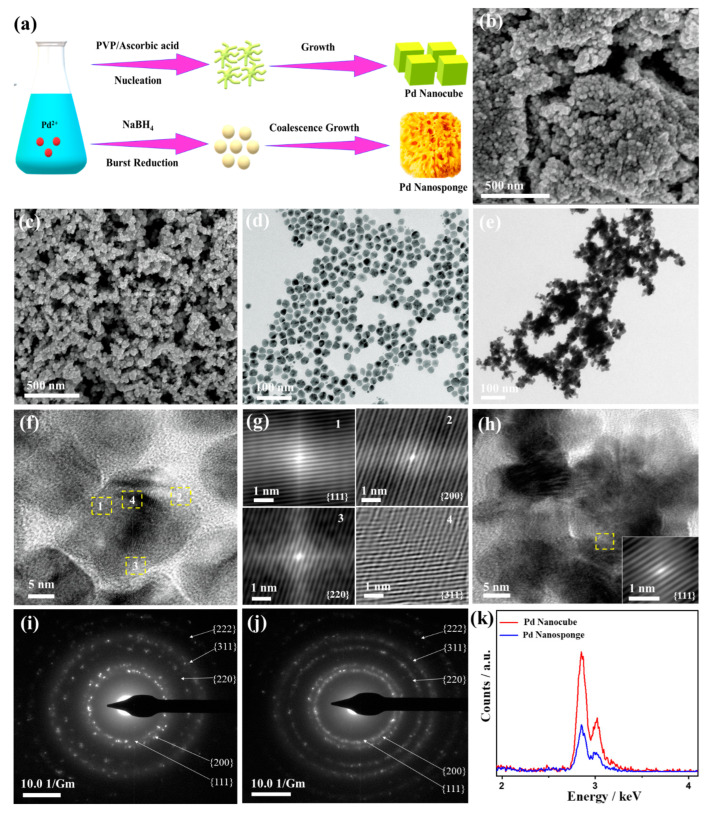
The formation process and mechanism of Pd nanocube and Pd nanosponge (**a**), SEM images (**b**,**c**), TEM images (**d**,**e**), HRTEM images (**f**,**h**), FFT images from marked spots (1–4) (**g**), SAED (**i**,**j**), and EDS analysis (**k**) of Pd nanocube and Pd nanosponge, respectively.

**Figure 2 ijms-24-11832-f002:**
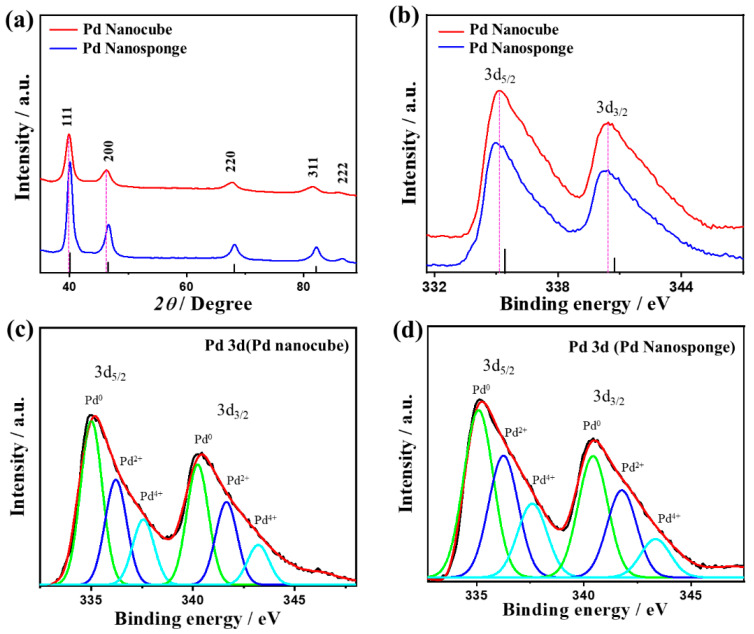
XRD (**a**), XPS full survey (**b**), and high-resolution Pd3d (**c**,**d**) of Pd nanocube and Pd nanosponge with color code: green (Pd^0^); blue (Pd^2+^) and cyan (Pd^4+^).

**Figure 3 ijms-24-11832-f003:**
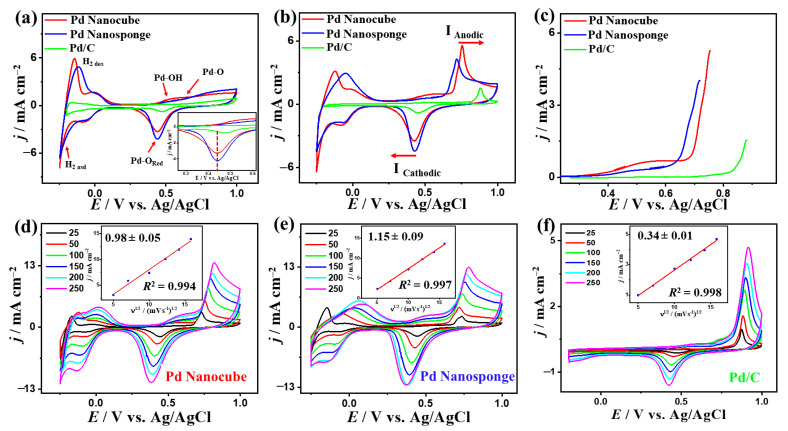
CV curves in N_2_-saturated 0.1 M H_2_SO_4_ (**a**), CO-saturated 0.1 M H_2_SO_4_ at 50 mV/s (**b**), LSV at 50 mV/s (**c**), and CV curves at different scan rates and their related plots of *I*_f_ vs. *υ*^1/2^ of Pd nanocube, Pd nanosponge, and Pd/C (**d**–**f**) in CO-saturated 0.1 M H_2_SO_4_.

**Figure 4 ijms-24-11832-f004:**
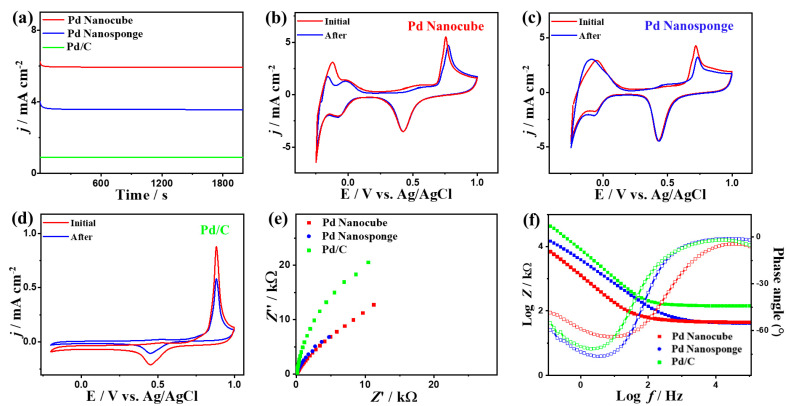
CA tests measured in CO-saturated 0.1 M H_2_SO_4_ (**a**), CV curves measured after CA (**b**–**d**), EIS (**e**), and Bode plots (**f**) of Pd nanocube, Pd nanosponge, and Pd/C.

**Figure 5 ijms-24-11832-f005:**
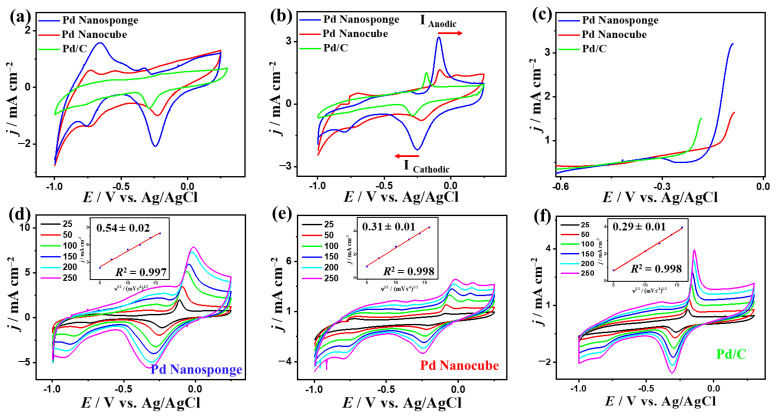
CV curves in N_2_-saturated 0.1 M KOH (**a**), CO-saturated 0.1 M KOH at 50 mV/s (**b**), LSV at 50 mV/s (**c**), and CV curves at different scan rates and their related plots of *I*_f_ vs. *υ*^1/2^ of Pd nanosponge, Pd nanocube, and Pd/C (**d**–**f**), in CO-saturated 0.1 M KOH.

**Figure 6 ijms-24-11832-f006:**
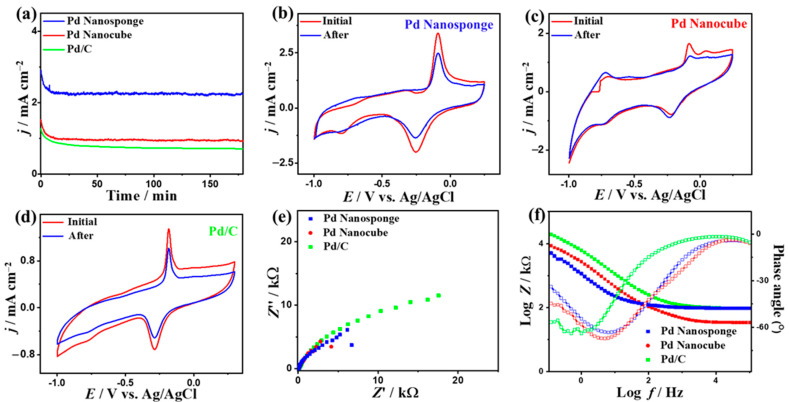
CA tests measured in CO-saturated 0.1 M KOH (**a**), CV curves measured after CA (**b**–**d**), EIS (**e**), and Bode plots (**f**) of Pd nanosponge, Pd nanocube, and Pd/C.

**Table 1 ijms-24-11832-t001:** Comparison of the electrochemical CO oxidation activity of Pd nanocube, Pd nanosponge, and Pd/C in different electrolytes.

Catalysts	*E*_Onset_ (V)	*E*_Oxid_ (V)	*I*_Anode_ mA cm^−2^ @ 50 mV/s	Slope (*I*_Anode_ vs. *υ* ^1/2^)	ECSA (m^2^/g)	Charge of COads (μC)	Mass/Specific Activity (mA·mg^−1^/mA·cm^−1^)
	Acid (0.1 M H_2_SO_4_)						
Pd Nanocube	0.360	0.755	5.92	0.92	22.88	930.0	19.85/0.090
Pd Nanosponge	0.410	0.715	4.00	1.25	23.64	955.5	14.13/0.060
Pd/C	0.750	0.878	1.59	0.34	11.11	265.0	18.7/0.095
	Alkaline (0.1 M KOH)						
Pd Nanocube	−0.133	−0.086	1.63	0.31	2.86	110.0	5.75/0.200
Pd Nanosponge	−0.170	−0.090	3.20	0.54	15.07	602.0	11.30/0.075
Pd/C	−0.217	−0.185	1.48	0.29	2.78	99.5	17.40/0.620
	Neutral (0.5 M NaHCO_3_)						
Pd Nanocube	0.200	0.371	0.578	0.196	6.23	130.0	2.04/0.065
Pd Nanosponge	−0.049	0.249	0.977	0.279	10.11	211.5	3.45/0.068
Pd/C	0.341	0.451	0.52	0.13	-3.16	64.4	6.12/0.110

## Data Availability

The data presented in this study are available on request from the corresponding authors.

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
