# Peer review of "Self-Standing Pd-Based Nanostructures for Electrocatalytic CO Oxidation: Do Nanocatalyst Shape and Electrolyte pH Matter?"

_ijms, 2023, doi:10.3390/ijms241411832_

Round 1
Reviewer 1 Report
· Please clarify the benefit of oxidizing CO to CO2 in the introduction. CO2 is a greenhouse gas which negatively affects the environment.
· Why didn’t the authors try to use support, especially for Pd nanosponge which may prevent nanoparticles’ aggregation? The support at least can decrease the aggregation.
· It is mentioned in lines 128-130 that ‘This is further evidenced in the lower lattice parameters of Pd nanocubes (3.20 Å) relative to Pd nanosponge (3.18 Å) and Pd/C (3.89 Å)’. The lattice parameters for Pd nanocubes and nanosponge are almost the same and even that of the former is slightly higher than that of the latter. Please correct this sentence.
· The authors mentioned that the onset oxidation potential on Pd nanocubes (0.715 V) is lower than that on Pd nanosponge (0.787), however, the CV curves (Fig. 3b, c) show the opposite case. Please clarify this point. Lines 177-180
· Please provide the CA curves at longer times and the corresponding CV curves after the new period. 2000 s (about 33 min) aren’t enough to judge the stability of the material.
· The authors concluded (lines 412-415) that the performance of Pd nanocubes is better than that of Pd nanosponge in all mediums which is opposite to what has been discussed in the manuscript where the performance of Pd nanocubes was only better in the acidic medium but inferior in the basic and neutral mediums.
· What is the Pd loading in the three catalysts? Is it 0.2831 µg for all or only for Pd nanocubes and Pd nanosponge?
· Please update the references to show the literature regarding the electrooxidation of CO. The references on this topic are very few on the references list.
The manuscript should be carefully checked grammatically.
Author Response
Date: June 22-2023
Dear Respected Reviewers,
We greatly appreciate the time and efforts the reviewers have dedicated to provide feedback on our manuscript "ijms-2438056” entitled "Self-standing Pd-based Nanostructures for the Electrocatalytic CO Oxidation: Do Nanocatalysts Shape and Electrolyte pH Matter ". We are grateful for the insightful comments and valuable improvements to our paper. We have incorporated all the suggestions made by the reviewers, with changes made highlighted in the manuscript in blue color. Please see appended below, in blue, a point-by-point response to the reviewers’ comments and concerns for your convenient evaluation.
Reviewer(s)' Comments to Author:
Reviewer 1
We thank Reviewer #1 for his/her critical and insightful comments on the paper, which significantly helped to improve the quality and clarity of this manuscript. We hope that our revisions and adaptations are adequate and reflect all the suggestions of Reviewer #1. Our detailed responses to reviewer #1 are given below.
Comment 1
Please clarify the benefit of oxidizing CO to CO2 in the introduction. CO2 is a greenhouse gas which negatively affects the environment.
Reply 1
Thank you for this comment. We have explained the benefit of oxidizing CO to CO2 in the introduction section as
“Among these, fuel cells operating with organic molecules (i.e., ethanol, glucose, ethylene glycol, and formic acid)[15-21] are among the most promising roadmaps due to their inferior emissions and green resources, but CO-poisoning of their anodes is one of the main barriers to preclude commercial usage [15,22-24]. So its important to develop electrocatalysts for efficient CO oxidation (COOxid) and also to optimize the oxidation process under different conditions that could allow protecting anodes of organic fuel cells from poisoning by CO. CO is highly poisoning gas and cause a lot of hazard effects on human health and environmental as well, so its conversion to less hazard CO2 gas is important. COOxid enables the mitigation of the hazardous effects and diversity of production sources of CO, with extensive applications in industries for the production of organic chemicals and environmental remediation[25-27].”
Comment 2
Why didn’t the authors try to use support, especially for Pd nanosponge which may prevent nanoparticles’ aggregation? The support at least can decrease the aggregation.
Reply 2
Many thanks for this great observation. Kindly allow us to emphasize that the use of support like carbon could makes the catalyst to be prone to detachment and aggregation, as in the case of commercial Pd/C inn this study. However, the crux of the study was to investigate the effect of shapes of free-standing ’’without Support’’ Pd catalysts and electrolytes’ pH for the electrochemical CO oxidation. On the other hand, we have compared the electrocatalytic COOxid of our self-standing Pd nanocube and Pd nanosponge with supported Pd/C and they were more stable against aggregation after the stability test than Pd/C
Comment 3
It is mentioned in lines 128-130 that ‘This is further evidenced in the lower lattice parameters of Pd nanocubes (3.20 Å) relative to Pd nanosponge (3.18 Å) and Pd/C (3.89 Å)’. The lattice parameters for Pd nanocubes and nanosponge are almost the same and even that of the former is slightly higher than that of the latter. Please correct this sentence.
Reply 3
Many thanks for this great observation. The sentence has been corrected as “This is further evidenced by the lower lattice parameters of Pd nanocube (3.20 Å) and Pd nanosponge (3.18 Å) relative to Pd/C (3.89 Å).”
Comment 4
The authors mentioned that the onset oxidation potential on Pd nanocubes (0.715 V) is lower than that on Pd nanosponge (0.787), however, the CV curves (Fig. 3b, c) show the opposite case. Please clarify this point. Lines 177-180
Reply 4
Many thanks for this great observation. The sentence has been corrected accordingly as “This is seen in the lower EOxid (EOnset) of Pd nanocube (0.76 V (0.36 V)) and Pd nanosponge (0.72 V (0.41 V)) relative to Pd/C (0.88 V (0.75 V)) (Fig. 3c).”
Comment 5
Please provide the CA curves at longer times and the corresponding CV curves after the new period. 2000 s (about 33 min) aren’t enough to judge the stability of the material.
Reply 5
Many thanks for this great observation. The CA curves have been repeated for 3 h and the corresponding CV curves after stability in the three electrolytes are shown in Figs. 4a, 6a and S6a.
Comment 6
The authors concluded (lines 412-415) that the performance of Pd nanocubes is better than that of Pd nanosponge in all mediums which is opposite to what has been discussed in the manuscript where the performance of Pd nanocubes was only better in the acidic medium but inferior in the basic and neutral mediums.
Reply 6
Many thanks for this great observation. The sentence has been corrected as “The electrocatalytic CO oxidation activity and durability of Pd nanocube were only significantly superior to Pd nanosponge and commercial Pd/C in acidic (H2SO4) medium and the best among the three media, due to the multiple adsorption active sites, uniform distribution, and high surface of nanocube structure. However, Pd nanosponge had enhanced COOxid activity and stability in both alkaline (KOH), and neutral (NaHCO3) electrolytes than Pd nanocube and Pd/C, attributable to its low Pd-Pd interatomic distance and cleaner surface.” However, more emphasis was laid on the COOxid in H2SO4 medium because of it best activity.
Comment 7
What is the Pd loading in the three catalysts? Is it 0.2831 µg for all or only for Pd nanocubes and Pd nanosponge?
Reply 7
Many thanks for this great observation. Yes, the Pd loading of 0.2831 µg was for the three catalysts.
Comment 8
Please update the references to show the literature regarding the electrooxidation of CO. The references on this topic are very few on the references list.
Reply 8
Many thanks for this great observation. 5 recent references relating to electrooxidation of CO have been added to the manuscript.
Comment 9
Comments on the Quality of English Language. The manuscript should be carefully checked grammatically.
Reply 9
Many thanks for this great observation. The quality of English Language and grammar have been carefully checked and corrected accordingly.
Reviewer 2 Report
I recommend to the authors a quick check of the manuscript with special attention to the abbreviations.
Author Response
Date: June 22-2023
Dear Respected Reviewers,
We greatly appreciate the time and efforts the reviewers have dedicated to provide feedback on our manuscript "ijms-2438056” entitled "Self-standing Pd-based Nanostructures for the Electrocatalytic CO Oxidation: Do Nanocatalysts Shape and Electrolyte pH Matter ". We are grateful for the insightful comments and valuable improvements to our paper. We have incorporated all the suggestions made by the reviewers, with changes made highlighted in the manuscript in blue color. Please see appended below, in blue, a point-by-point response to the reviewers’ comments and concerns for your convenient evaluation.
Reviewer(s)' Comments to Author:
Reviewer 2
We thank Reviewer #2 for finding our work worthy of being published in the Journal of Molecular Science. We have thoroughly checked the abbreviation and corrected them accordingly, which significantly helped to improve the quality and clarity of this manuscript. We hope that our revisions and adaptations are adequate and reflect all the suggestions of Reviewer #2. Our detailed responses to reviewer #2 are given below.
Comment
I recommend to the authors a quick check of the manuscript with special attention to the abbreviations.
We have revised all the abbreviations in the revised manuscript
Reviewer 3 Report
This article presents the controlled fabrication of Pd in different morphologies, including nanosponge by ice-cooling reduction of Pd- precursors by NaBH4 solution Pd nanocubes by ascorbic acid reduction at 25oC. Both Pd nanosponge and Pd nanocubes are self-standing and possess high surface area, uniform distribution, and clean surface.
What parameters can be additionally be adjusted specifically by used methods for efficient CO oxidation to which products (?) and stability achieved in testing need to be defined in abstract and elsewhere? Regeneration of used materials needs to be shown. Necessity for doping needs to be outlined.
Benefits of Pd over Pt needs to be outlined as both of them are expensive and alternatives are available for CO oxidation, elaborate.
English quality needs to be adjusted significantly in current MS as well as to give detailed methods description.
Following questions have arisen:
- The title should say something about novel result of the research and show the innovative result. This application is not definitely increasing the efficiency of the system alone when not tested for different configurations and compared with other measurements by You and in comparison with latest literature. Limit the number of figures and tables, give only the most important one’s results. Error bars are mandatory in figures.
- The objective of this study is needed to quantify the extent to which CO oxidation occurred, which needs justification as well as to make less broader objectives and study scientifically more relevant. Figure 1.- all subfigures need to be explained, such as i) J, which of some are very similar and can be neglected: i), j)
- R84. „Schema1 describes the synthesis pathways of Pd nanocubes nanostructures via …“- Should be Fig. 1
- R193 „However, the line slope of Pd nanosponges (0.98) and Pd nanocubes (0.99) was significantly greater than that of Pd/C (0.198)- it was not significant, what was p value to state significance in Your results? Figure 3. Regression coefficient sign R2 is missing from Center correlation figures. „Figure 3. CVs in N2-staruraed“- should be „saturated“.
- R205 “while porous Pd nanosponge allows quick diffusion of reactants and tolerate the adsorption of intermediates[12,38-40]..“-remove one full stop. Check latest literature on equilibrium processes affecting adsoption and treatment effect of wastewater: https://doi.org/10.1002/er.8300 https://doi.org/10.3176/proc.2018.3.04 https://doi.org/10.1080/09593330.2011.588962, https://doi.org/10.3176/proc.2018.3.10, https://doi.org/10.1021/acs.jpca.7b00237,
- Unify Figs subscripts , units as mV not mv, color legend, legends in Figs and make data difference visible well, error bars are necessary to add. Graphical abstract is missing. Chemical results are needed to be shown first, then showing characterization techniques in MS.
Needs revision
Author Response
Date: June 22-2023
Dear Respected Reviewers,
We greatly appreciate the time and efforts the reviewers have dedicated to provide feedback on our manuscript "ijms-2438056” entitled "Self-standing Pd-based Nanostructures for the Electrocatalytic CO Oxidation: Do Nanocatalysts Shape and Electrolyte pH Matter ". We are grateful for the insightful comments and valuable improvements to our paper. We have incorporated all the suggestions made by the reviewers, with changes made highlighted in the manuscript in blue color. Please see appended below, in blue, a point-by-point response to the reviewers’ comments and concerns for your convenient evaluation.
Reviewer(s)' Comments to Author:
Reviewer 3
Comments:
This article presents the controlled fabrication of Pd in different morphologies, including nanosponge by
ice-cooling reduction of Pd- precursors by NaBH4 solution Pd nanocubes by ascorbic acid reduction at
25 oC. Both Pd nanosponge and Pd nanocubes are self-standing and possess high surface area, uniform
distribution, and clean surface.
We thank Reviewer #3 for his/her critical and insightful comments on the paper, which significantly helped to improve the quality and clarity of this manuscript. We hope that our revisions and adaptations are adequate and reflect all the suggestions of Reviewer #3. Our detailed responses to reviewer #3 are given below.
Comment 1
What parameters can be additionally be adjusted specifically by used methods for efficient CO oxidation to which products (?) and stability achieved in testing need to be defined in abstract and elsewhere? Regeneration of used materials needs to be shown. Necessity for doping needs to be outlined.
Reply 1
Thanks for the great comments.
We have deeply studied and optimized the electrocatalytic COOxid as a function of support-free Pd shape and electrolyte pH compared with supported Pd/C catalyst which are the main aims of this study. The results clearly warranted the superior activity of self-standing Pd with anisotropic shapes (nanocubes and nanosponge) in all electrolytes than supported Pd/C. The additional parameter could be the electrode type but it can not make a significant contribution because we have used metal Pd catalyst not carbon ones.
The stability of the Pd nanocatalysts has been explained briefed in the abstract, introduction and conclusion.
The regeneration is shown in Figs. S4b-S4d and explained as “Moreover, the Pd nanocrystals and Pd/C were regenerated after CO oxidation in 0.1 M H2SO4 electrolyte by purging the solution with N2 for 1 h to remove the adsorbed CO and/or CO2 on the catalysts’ active sites (Figs. S4b-S4d)”.
Kindly allow us to emphasize that doping is not the confinement of the crux of this study and thus not discussed. This work only focus on the effect of Pd shapes and electrolyte pH for electrochemical CO oxidation.
Comment 2
Benefits of Pd over Pt needs to be outlined as both of them are expensive and alternatives are available for CO oxidation, elaborate.
Reply 2
Thanks for your careful reading. Kindly allow us to emphasize that, although the high cost and earth rarity of Pd and Pt, they are still the best commercial electrocatalysts for fuel cells and also for the electrochemical CO oxidation at lower overpotential, so optimization their CO oxidation performance is of great importance in various fundamental, energy, and industrial applications.
Comment 3
English quality needs to be adjusted significantly in current MS as well as to give detailed methods description.
Reply 3
Thanks for your careful reading. The quality of English in the manuscript has been corrected accordingly.
Following questions have arisen:
Question 1
The title should say something about novel result of the research and show the innovative result. This application is not definitely increasing the efficiency of the system alone when not tested for different configurations and compared with other measurements by You and in comparison with latest literature. Limit the number of figures and tables, give only the most important one’s results. Error bars are mandatory in figures.
Reply 1
Thanks for your careful reading and this observation. The title with the rhetoric question has captured the effect of catalyst shapes and electrolytes’ pH play important roles in improving the electrochemical CO oxidation.
The performance of the as-prepared Pd nanocatalysts in this work is compared in different catalysts for electrochemical CO oxidation in Table S2.
To limit the number of Figs. and tables, we have moved Figs. 7 and 8 to supporting information as Figs. S5 and S6 and Tables 2-4 as Table S3-S5.
Error bars have been added to Figs. 3d-3f, 5d-4f and S5d-S5f.
Question 2
The objective of this study is needed to quantify the extent to which CO oxidation occurred, which needs justification as well as to make less broader objectives and study scientifically more relevant. Figure 1.- all subfigures need to be explained, such as i) J, which of some are very similar and can be neglected: i), j)
Reply 2
Thank you for these comments. Kindly allow us to emphasize that the quantification of CO oxidation occurrence was explained in terms of current density of Pd nanocube, Pd nanospsonge and Pd/C in the Abstract and Conclusion sections. Also, the amount of CO is justified corresponding to the charge coverage in the three electrolytes is highlighted in Table 1 and discussed as “the amount of CO adsorbed on Pd nanocube (930 μC) and Pd nanosponge (955.5 μC) and Pd/C (265 μC) in 0.1 M H2SO4, as obtained from integrated CO oxidation.”
The subfigures in Figs. 1i, 1j have been properly explained.
Question 3
R84. „Schema1 describes the synthesis pathways of Pd nanocubes nanostructures via …“- Should be Fig. 1.
Reply 3
Thank you for this observation. “Scheme 1” has been replaced with “Fig. 1a”
Question 4
R193 „However, the line slope of Pd nanosponges (0.98) and Pd nanocubes (0.99) was significantly greater than that of Pd/C (0.198)- it was not significant, what was p value to state significance in Your results? Figure 3. Regression coefficient sign R2 is missing from Center correlation figures. „Figure 3. CVs in N2-staruraed“- should be “saturated“.
Reply 4
Thank you for these comments. The statement has been reworded as “However, the line slopes of Pd nanosponges (0.98 ± 0.05) and Pd nanocubes (1.15 ± 0.09) were higher than that of Pd/C (0.34 ± 0.01) with regression coefficients (0.994-0.998) (Figs. 3d-3f)…”
The regression coefficients have been added to Figs. 3d-3f, 5d-5f and S5d-S5f.
“Fig. 3. CVs in N2-staruraed…” is corrected as “Fig. 3. CVs in N2-saturated…”
Question 5
R205 “while porous Pd nanosponge allows quick diffusion of reactants and tolerate the adsorption of intermediates[12,38-40]..“-remove one full stop. Check latest literature on equilibrium processes affecting adsoption and treatment effect of wastewater: https://doi.org/10.1002/er.8300 https://doi.org/10.3176/proc.2018.3.04 https://doi.org/10.1080/09593330.2011.588962, https://doi.org/10.3176/proc.2018.3.10, https://doi.org/10.1021/acs.jpca.7b00237,
Reply 5
Thank you for this observation. The statement has been corrected as “while porous Pd nanosponge allows quick diffusion of reactants and tolerate the adsorption of intermediates[12,43-45].” The recent literature have been cited.
Question 6
Unify Figs subscripts , units as mV not mv, color legend, legends in Figs and make data difference visible well, error bars are necessary to add. Graphical abstract is missing. Chemical results are needed to be shown first, then showing characterization techniques in MS.
Reply 6
Thank you for these comments. The Figs. have been unified, whereas units and color legends in the Figs. are corrected, and the data visible. We have also added the error bars where necessary. Graphical abstract is added on the last page of the MS after references. The MS is well arranged according to the Journal of Molecular Science strict guidelines.
Question 7
Comments on the Quality of English Language
Reply 7
Thank you for these comments. The quality of English Language has been improved.
Round 2
Reviewer 1 Report
The reviewer can't believe the results obtained by the authors. The data after the CA test for 33 min reported in the first version of this manuscript are the same as those obtained after 3 h. It looks like the authors didn't do the test for 3 h, instead they multiple the time by a factor like 5 or 6. In this regard, I don't think that the authors did real tests for this study.
I strongly recommend the editor to reject this manuscript
The manuscript needs moderate revision for English.
Author Response
Date: July 11-2023
Dear Respected Reviewers,
We greatly appreciate the time and efforts the reviewers have dedicated to provide feedback on our manuscript "ijms-2438056” entitled "Self-standing Pd-based Nanostructures for the Electrocatalytic CO Oxidation: Do Nanocatalysts Shape and Electrolyte pH Matter ". We are grateful for the insightful comments and valuable improvements to our paper. We have incorporated all the suggestions made by the reviewers, with changes made highlighted in the manuscript in blue color. Please see appended below, in blue, a point-by-point response to the reviewers’ comments and concerns for your convenient evaluation.
Reviewer(s)' Comments to Author:
Reviewer 1
Comment
The reviewer can't believe the results obtained by the authors. The data after the CA test for 33 min reported in the first version of this manuscript are the same as those obtained after 3 h. It looks like the authors didn't do the test for 3 h, instead they multiple the time by a factor like 5 or 6. In this regard, I don't think that the authors did real tests for this study.
I strongly recommend the editor to reject this manuscript''.
We thank Reviewer #1 for his/her great comments, which has helped to improve the quality and clarity of the manuscript. Kindly allow us to emphasize the data curation was carefully obtained. The degradation of catalysts in CA curves only occur after first-few minutes before stabilization. We have replaced the CA curves with new ones (Please see Figs. 4a, 6a and S6a). All the experiments were rigorously carried out with optimal accuracy. We hope that our revisions and adaptations are adequate and reflect all the suggestions of Reviewer #1. We appeal to reviewer #1 to reconsider his/her decision for acceptance of the manuscript in the International Journal of Molecular Science.
Round 3
Reviewer 1 Report
The authors improved the quality of their manuscript. I accept the revised manuscript in its current form.